# Fluorescence Lifetime Imaging Microscopy reveals rerouting of SNARE trafficking driving dendritic cell activation

**Daniëlle Rianne José Verboogen, Natalia González Mancha, Martin ter Beest, Geert van den Bogaart***

Department of Tumor Immunology, Radboud Institute for Molecular Life Sciences, Radboud University Medical Center, Nijmegen, Netherlands

**Abstract** SNARE proteins play a crucial role in intracellular trafficking by catalyzing membrane fusion, but assigning SNAREs to specific intracellular transport routes is challenging with current techniques. We developed a novel Förster resonance energy transfer-fluorescence lifetime imaging microscopy (FRET-FLIM)-based technique allowing visualization of real-time local interactions of fluorescently tagged SNARE proteins in live cells. We used FRET-FLIM to delineate the trafficking steps underlying the release of the inflammatory cytokine interleukin-6 (IL-6) from human blood-derived dendritic cells. We found that activation of dendritic cells by bacterial lipopolysaccharide leads to increased FRET of fluorescently labeled syntaxin 4 with VAMP3 specifically at the plasma membrane, indicating increased SNARE complex formation, whereas FRET with other tested SNAREs was unaltered. Our results revealed that SNARE complexing is a key regulatory step for cytokine production by immune cells and prove the applicability of FRET-FLIM for visualizing SNARE complexes in live cells with subcellular spatial resolution.

*For correspondence: geert. vandenbogaart@radboudumc.nl

**Competing interests:** The authors declare that no competing interests exist.

## Introduction

One of the central paradigms in cell biology is that all intracellular membrane fusion, except for mitochondrial fusion, is catalyzed by soluble NSF (N-ethylmaleimide-sensitive fusion protein) attachment protein receptor (SNARE) proteins (*Hong, 2005*; *Jahn and Scheller, 2006*). There are about 36 SNARE proteins identified in mammals which can be classified in R-SNAREs or Q-SNAREs depending on the central residue of their SNARE motif being either arginine (R) or glutamine (Q). The interaction of one R motif and three Q motifs (termed Qa, Qb and Qc) of cognate SNARE proteins located in opposing membranes leads to the formation of a 4-helix coiled-coil bundle. This *trans*-SNARE complex formation brings the membranes in close proximity and catalyzes their fusion. After membrane fusion is complete, the SNAREs are in a so-called *cis*-conformation and need to be disassembled by the AAA-ATPase NSF in conjunction with α-SNAP (*Söllner et al., 1993*; *Hong, 2005*; *Jahn and Scheller, 2006*).

From studies in yeast and many mammalian cell types, including neurons, neuroendocrine cells, adipocytes and epithelial cells, it is clear that different intracellular trafficking steps are catalyzed by specific sets of SNARE proteins (*Pelham, 2001*; *Hong, 2005*; *Jahn and Scheller, 2006*). Also in immune cells, such as granulocytes, platelets, mast cells, macrophages and T cells, specific combinations of SNARE proteins mediate release of cytokines, chemokines and delivery of surface receptors via specific secretory pathways (*Collins et al., 2015*; *Stow et al., 2009*; *Stanley and Lacy, 2010*; *Lacy and Stow, 2011*; *Murray and Stow, 2014*). In order to identify the combinations of SNARE proteins responsible for a particular intracellular trafficking step, most studies rely on localization microscopy experiments either with endogenous or overexpressed SNAREs. However, these

**eLife digest** Many processes in living cells involve membranes coming together and fusing. For example, white blood cells known as dendritic cells rely on membrane fusion to fight off infections. When a dendritic cell detects a bacterial infection, it releases signaling molecules called cytokines to recruit other immune cells that help to eliminate the bacteria. The cytokines are contained in membrane-bound packages inside the cell, called vesicles, and are transported outside when these vesicles fuse with the membrane that surrounds the dendritic cell.

Proteins called SNAREs drive the fusion of a cell's membranes. These proteins, which are found on both membranes that will fuse, entwine to form a tight complex that pulls the membranes together. Mammals have over 30 different SNARE proteins, and many scientists believe that specific transport routes within cells use distinct pairs of SNAREs. However, to date, it has been difficult to assign specific pairs of SNAREs to specific transport routes with existing techniques.

Verboogen et al. have now engineered human dendritic cells to add labels onto their SNAREs that fluoresce if the proteins interact. This approach meant that the interactions could be tracked via a microscope. The experiments showed that exposing dendritic cells to a bacterial compound that stimulates the release of cytokines caused two SNARE proteins called syntaxin 4 and VAMP3 to interact more at the cell membrane. This indicates that syntaxin 4 and VAMP3 are important for the release of cytokines from these cells.

This finding was supported by an additional experiment in which Verboogen et al. switched off the gene for VAMP3 in the dendritic cells and found that this reduced the amount of cytokines that were released. This new microscope-based approach will be useful for identifying the specific pairs of SNARE proteins that are needed for the release and transport of molecules – like hormones and enzymes – that are important in health and disease.

approaches suffer from the problems that many SNAREs locate to multiple organelles, as SNAREs are promiscuous and can be involved in different trafficking steps, and that mere co-localization of SNAREs does not prove their actual interaction (*Pelham, 2001*; *Hong, 2005*). Interactions can be shown with co-immunoprecipitation experiments, but these cannot resolve in which organelle the interactions take place. Perturbation experiments, such as genetic ablation of SNAREs, overexpression of soluble 'dominant negative' SNARE fragments, or microinjection of antibodies directed against SNAREs, are also difficult to interpret because SNAREs are functionally redundant and defects in a specific trafficking route cannot be discerned from upstream trafficking steps. Because of this, knockdown or knockout of many SNARE proteins often does not show a clear phenotype (*Hong, 2005*; *Bethani et al., 2009*; *Nair-Gupta et al., 2014b*). Thus, a method that allows quantitative visualization of SNARE complexes with subcellular resolution is highly desirable.

In this study, we aimed to fill this gap in methodology by developing an assay to locally detect SNARE interactions with Förster resonance energy transfer (FRET) based fluorescence lifetime imaging microscopy (FLIM) (*Jares-Erijman and Jovin, 2003*; *Wallrabe and Periasamy, 2005*). FRET is based on the energy transfer from a donor to an acceptor fluorophore within the Förster distance range. FRET leads to quenching of the donor fluorophore and sensitization of the acceptor fluorophore and can either be measured from fluorescence intensities (ratiometric FRET) or from a decrease in fluorescence lifetime of the donor fluorophore (FRET-FLIM). Ratiometric FRET and FRET-FLIM have been previously used to characterize cytosolic interactions of fungal SNAREs (*Valkonen et al., 2007*) and of the neuronal Qa-SNARE syntaxin 1 (Stx1) and the Qb/c-SNARE SNAP25 (*Rickman et al., 2007*; *Medine et al., 2007*; *Rickman et al., 2010*; *Xia et al., 2001*; *Kavanagh et al., 2014*). We hypothesized that FRET-FLIM could also be used to measure full ternary *cis*-SNARE complex formation, as the crystal structure of the neuronal SNARE complex revealed that the luminal/extracellular C-termini of the R-SNARE vesicle-associated membrane protein 2 (VAMP2) and of Stx1 are in immediate proximity (<1 nm) after membrane fusion (i.e., in the *cis*-SNARE complex) (*Stein et al., 2009*). In vitro, FRET between these SNAREs can be measured by labeling their C-termini by means of site-specific labeling with organic fluorophores (*Xia et al., 2001*). In vivo, *Degtyar et al. (2013)* showed interactions between VAMP2 and Stx1 C-terminally fused to

fluorescent proteins by total internal reflection fluorescence (TIRF) microscopy combined with ratiometric FRET (*Degtyar et al., 2013*). However, ratiometric FRET is technically challenging as extensive controls are required to correct for local concentration differences of the donor and acceptor fluorophores (*Jares-Erijman and Jovin, 2003*; *Wallrabe and Periasamy, 2005*). FLIM does not suffer from this limitation, as the lifetime (τ) is an intrinsic characteristic of a fluorophore which is not influenced by the probe concentration nor by the excitation intensity (*Wallrabe and Periasamy, 2005*; *Jares-Erijman and Jovin, 2003*). Occurrence of FRET leads to quenching of the donor signal, which in turn shortens its fluorescence lifetime. Therefore, reductions in the donor's lifetime reflect interactions between the proteins conjugated to the donor and acceptor fluorophores. In this study, we demonstrate the applicability of our FRET-FLIM assay to visualize the complexes of several SNAREs in dendritic cells of the immune system.

Dendritic cells are leukocytes essential for the activation of T cells (*Banchereau and Steinman, 1998*). Activation of dendritic cells by inflammatory or pathogenic stimuli triggers the production and release of cytokines and chemokines that orchestrate the immune response (*Collins et al., 2015*; *Stanley and Lacy, 2010*). Despite many studies characterizing the cytokines that are released by dendritic cells and their mechanisms of action, not much is known about the trafficking pathways involved. Using FRET-FLIM, we measured how activation of dendritic cells affects the interactions between the R-SNAREs vesicle-associated membrane protein 3 (VAMP3) and VAMP8 with the Qa-SNAREs syntaxin 3 (Stx3) and syntaxin 4 (Stx4). While both VAMP3 and VAMP8 are associated with cytokine release from immune cells (*Collins et al., 2015*; *Stow et al., 2009*; *Stanley and Lacy, 2010*; *Lacy and Stow, 2011*; *Murray and Stow, 2014*), VAMP3 is mostly associated with early and recycling endosomes and VAMP8 with late endosomes (*Bajno et al., 2000*; *Manderson et al., 2007*; *Hong, 2005*; *Murray et al., 2005*; *Antonin et al., 2000*). VAMP8 has also been reported to inhibit phagocytosis by dendritic cells (*Ho et al., 2008*). Stx3 and Stx4 are mainly plasma membrane localized SNAREs mediating exocytosis, but also have intracellular roles in immune cells (*Naegelen et al., 2015*; *Collins et al., 2014*; *Stow et al., 2009*; *Stanley and Lacy, 2010*; *Lacy and Stow, 2011*; *Murray and Stow, 2014*; *Gómez-Jaramillo et al., 2014*; *Frank et al., 2011*). Our FRET-FLIM experiments revealed FRET of fluorescently labeled Stx3 with VAMP3 in live dendritic cells and this was predominantly present at the plasma membrane, indicative of increased Stx3-VAMP3 complexing at this site. In contrast, FRET of fluorescently labeled Stx3 with VAMP8 was higher at intracellular compartments. Our results also showed an increase in FRET of fluorescently labeled Stx4, but not of Stx3, with VAMP3 upon cellular activation with the Toll-like receptor 4 (TLR4) agonist lipopolysaccharide (LPS), indicating that LPS promotes the complex formation of Stx4 with VAMP3. LPS activation of dendritic cells induces an inflammatory response, which is characterized by the secretion of inflammatory cytokines such as interleukin-6 (IL-6) (*Stow et al., 2009*). VAMP3 plays a key role in this cytokine secretion, as siRNA knockdown of VAMP3 resulted in impaired release of IL-6 from dendritic cells. Our study demonstrates that the secretion of IL-6 is regulated at the level of SNARE complex formation and proves the use of FRET-FLIM for quantitative visualization of SNARE interactions in live cells.

## Results

### SNAREs interactions in live cells visualized by FLIM

We generated variants of VAMP3 and Stx3 with their C-termini conjugated to mCitrine (*Griesbeck et al., 2001*) and mCherry (*Shaner et al., 2004*), respectively. The emission spectrum of mCitrine overlaps with the excitation spectrum of mCherry which makes them a suitable donor-acceptor pair for FRET (*Figure 1—figure supplement 1A*). Both mCitrine and mCherry are pH-insensitive and therefore their fluorescence lifetimes are not affected by low pH within the lumen of intracellular compartments (*Griesbeck et al., 2001*; *Shaner et al., 2004*). In addition, they are monomeric which is important to prevent oligomerization artifacts. Modeling the crystal structure of the neuronal *cis*-SNARE complex (*Stein et al., 2009*) with mCitrine (*Ho et al., 2008*) and mCherry (*Shu et al., 2006*) fused to C-termini of VAMP2 and Stx1 showed that these fluorophores are immediately juxtaposed to each other (*Figure 1A*). Based on the high structural homology of SNARE proteins, we expected that mCitrine and mCherry attached to the C-termini of VAMP3 or VAMP8 and

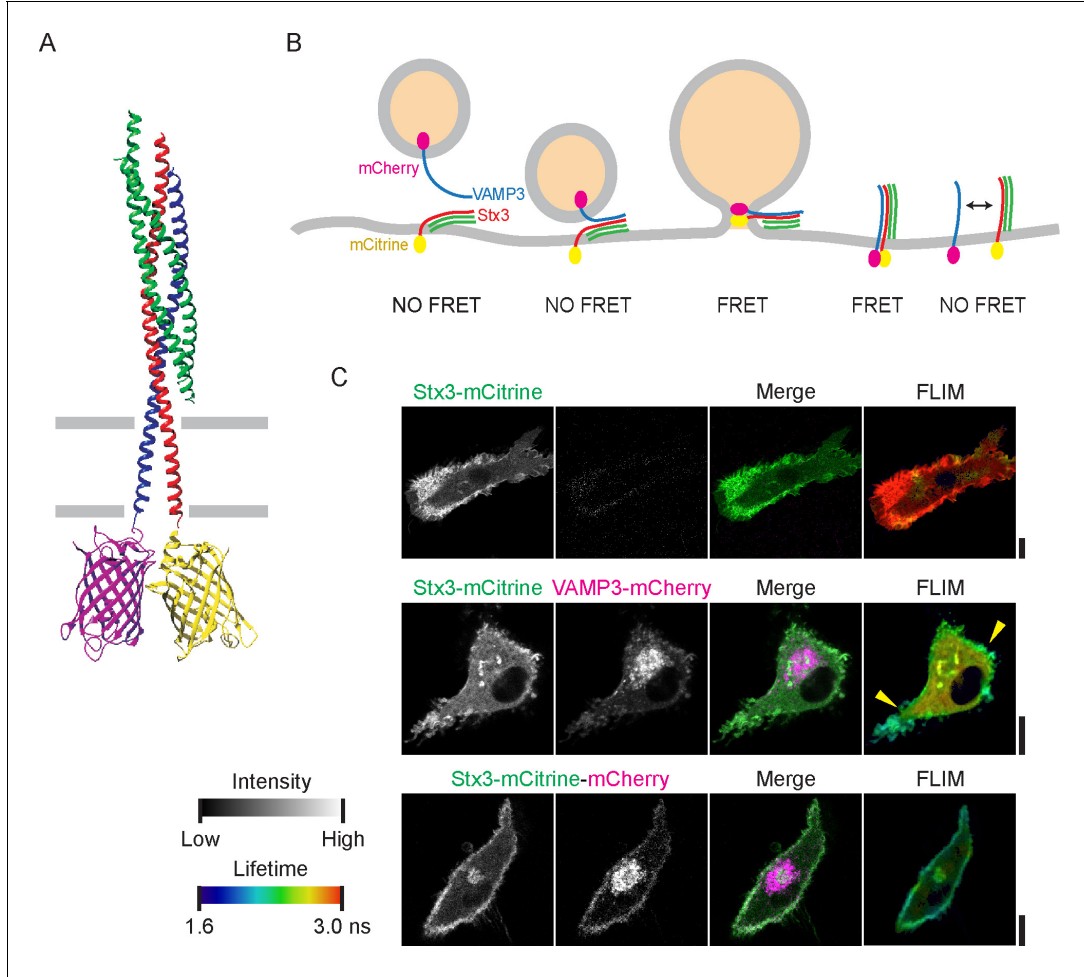

**Figure 1.** SNARE complex formation by FRET-FLIM. (**A**) Model of the neuronal SNAREs (crystal structure; protein database 3HD7 [**Stein et al., 2009**]) with the C-termini of syntaxin 1 (red) conjugated to mCitrine (3DQ1 [**Ho et al., 2008**]; yellow) and VAMP2 (blue) conjugated to mCherry (2H5Q [**Shu et al., 2006**]; magenta). mCitrine (donor fluorophore) and mCherry (acceptor) are within 3 nm proximity resulting in FRET. Green: SNAP25. (**B**) Scheme of membrane fusion resulting in FRET. (**C**) Representative confocal microscopy (left) and FLIM (right) images of dendritic cells expressing Stx3-mCitrine (green in merge; upper panels), Stx3-mCitrine with VAMP3-mCherry (magenta; middle panels), or Stx3 conjugated to both mCitrine and mCherry (Stx3-mCitrine-mCherry; lower panels). Apparent fluorescence lifetimes of Stx3-mCitrine with VAMP3-mCherry were lowest at the cell membrane (yellow arrowheads). Scale bars, 10 µm. Full lifetime/intensity lookup table and lifetime images are in *Figure 1—figure supplement 1D–E*.

The following figure supplement is available for figure 1:

**Figure supplement 1.** Lifetime images, confocal images and phasor analysis belonging to main *Figure 1*.

Stx3 or Stx4 would also be in close proximity in a *cis*-SNARE complex and that this would result in FRET and a decrease of the donor (mCitrine) fluorescence lifetime (*Figure 1B*).

Human blood-isolated monocyte-derived dendritic cells were transfected with either Stx3-mCitrine alone or in combination with VAMP3-mCherry (*Figure 1C*). Stx3-mCitrine localized to the plasma membrane and to intracellular compartments. VAMP3-mCherry was predominantly present in intracellular compartments with only a small fraction located at the plasma membrane similar to the localization of endogenous VAMP3 (*Figure 1—figure supplement 1B*). We recorded fluorescent lifetime images of live (unfixed) cells at an image plane locating at the height of the nucleus and accumulated on average ~750,000 photons per cell (*Figure 1—figure supplement 1C*). The fluorescence life time histograms of single pixels were fitted with a mono-exponential decay function to generate FLIM images showing the apparent lifetime of each pixel (*Figure 1C*; *Figure 1—figure supplement 1D–E*). Compared to cells expressing only Stx3-mCitrine, the FLIM images of cells co-

transfected with Stx3-mCitrine and VAMP3-mCherry showed a reduction of the donor's apparent lifetime. The lowest apparent lifetime was present at the plasma membrane (quantification below), indicating that the complexing of VAMP3 with Stx3 mainly took place at the cell membrane and less at intracellular compartments. FRET was also clear from Phasor analysis (*Hinde et al., 2012*), as co-expression of Stx3-mCitrine with VAMP3-mCherry resulted in a shift of the phasor location compared to the condition with the donor fluorophore only (Stx3-mCitrine) (*Figure 1—figure supplement 1F*). Thus, FRET-FLIM allows visualization of SNARE complexes in living cells.

In order to obtain an estimate of the lowest lifetime observable, we recorded FLIM images of Stx3 conjugated to both mCitrine and mCherry in tandem (Stx3-mCitrine-mCherry) (*Figure 1C*; *Figure 1—figure supplement 1E–F*). Although the colocalization of the mCitrine and mCherry signals was apparent, the overlap was not complete and the mCherry signal was more present in a cellular area juxtaposed to the nucleus (*Figure 1C*; *Figure 1—figure supplement 1G–H*). To investigate this phenomenon further, we co-expressed VAMP3-mCitrine with VAMP3-mCherry or VAMP8-mCitrine with VAMP8-mCherry and compared the localization of these constructs (*Figure 1—figure supplement 1I*). Also in this case, the mCherry signals were more prevalent than mCitrine in a juxtanuclear area. This juxtanuclear accumulation of mCherry was also observed upon dissipating the proton gradients by paraformaldehyde fixation (*Figure 1—figure supplement 1G and J*), indicating that the differential localization of mCherry and mCitrine was not caused by pH-quenching of mCitrine, but possibly due to differences in maturation speed or stability of the used fluorophores (*Griesbeck et al., 2001*; *Shaner et al., 2004*). Consequently, our FRET-FLIM approach likely underestimates the amount of FRET in juxtanuclear regions and more stable or rapidly maturating YFP analogs need to be developed for this. However, we observed clear overlap of mCherry and mCitrine in more peripheral cellular regions (Pearson correlation coefficients between 0.6–0.8 for the whole cell; *Figure 1—figure supplement 1H and K*), supporting the conclusion that our FRET-FLIM method can report on SNARE interactions in those regions.

We performed whole-cell fluorescence lifetime analysis to directly compare between cells and conditions. We fitted all pooled photons collected from each individual cell with single exponential decay functions convoluted with the instrument response function (IRF) (*Figure 2A–B*; *Figure 2—figure supplement 1A–B*). We obtained reasonable fitting accuracy (within ~2% deviation), although these deviations were larger at very short time intervals (<2.5 ns; *Figure 2—figure supplement 1A*). These deviations at short time intervals are likely caused by imperfect fitting of the IRF due to drift of the laser pulsing or timing of the detectors, reflections and/or (auto)fluorescence with fast kinetics, but do not cause major deviations of the resulting apparent lifetimes as our experiments with the reference dye rhodamine B show (*Figure 2—figure supplement 1C*). In Stx3-mCitrine expressing cells, co-transfection with VAMP3-mCherry significantly reduced the apparent lifetime of mCitrine from 2.79 ± 0.02 ns to 2.49 ± 0.03 ns (mean ± SEM) (*Figure 2C*). Cells from at least four different donors (>8 cells/donor) were measured for each condition (*Figure 2—figure supplement 1D*). The spread of apparent lifetimes for cells co-expressing Stx3-mCitrine with VAMP3-mCherry was quite large, ranging from 2 to 3 ns (*Figure 2C*). This large spread was at least partly caused by the availability of VAMP3-mCherry and competition with endogenous SNAREs, because the fluorescence lifetimes inversely correlated with the expression levels of VAMP3-mCherry (*Figure 2D*; *Figure 2—figure supplement 1E*). To account for the whole cell population with varying levels of acceptor fluorophore-labeled SNAREs, we randomly selected cells with visible expression of both donor and acceptor fluorophores for our FRET-FLIM experiments. In order to quantify the difference in apparent lifetimes between the plasma membrane and intracellular compartments, we manually selected peripheral and intracellular regions of the image cell areas and fitted the fluorescence lifetime histograms of these two areas with mono-exponential decay functions (*Figure 2—figure supplement 1F–G*). Most cells showed a lower apparent lifetime at the cellular periphery compared to the intracellular region (*Figure 2E*). This reduction was on average 55 ps (paired two-sided Student's $t$-test; p=0.0078), but was larger for cells with higher FRET (linear regression: $\beta$ = 1.18, $R^2$ = 0.82).

The co-expression of Stx3-mCitrine and VAMP3-mCherry will result in a mixed population, with part of Stx3-mCitrine interacting with VAMP3-mCherry and the remainder free or engaged with endogenous VAMP3 or other R SNAREs. In order to estimate the fraction of FRET, we fitted the whole-cell lifetime histograms of cells co-expressing Stx3-mCitrine and VAMP3-mCherry with double exponential decay functions (*Figure 2F*; *Figure 2—figure supplement 2A*). In our biexponential fitting, we had to fix the lifetime components, as we did not have sufficient photon counts for extra

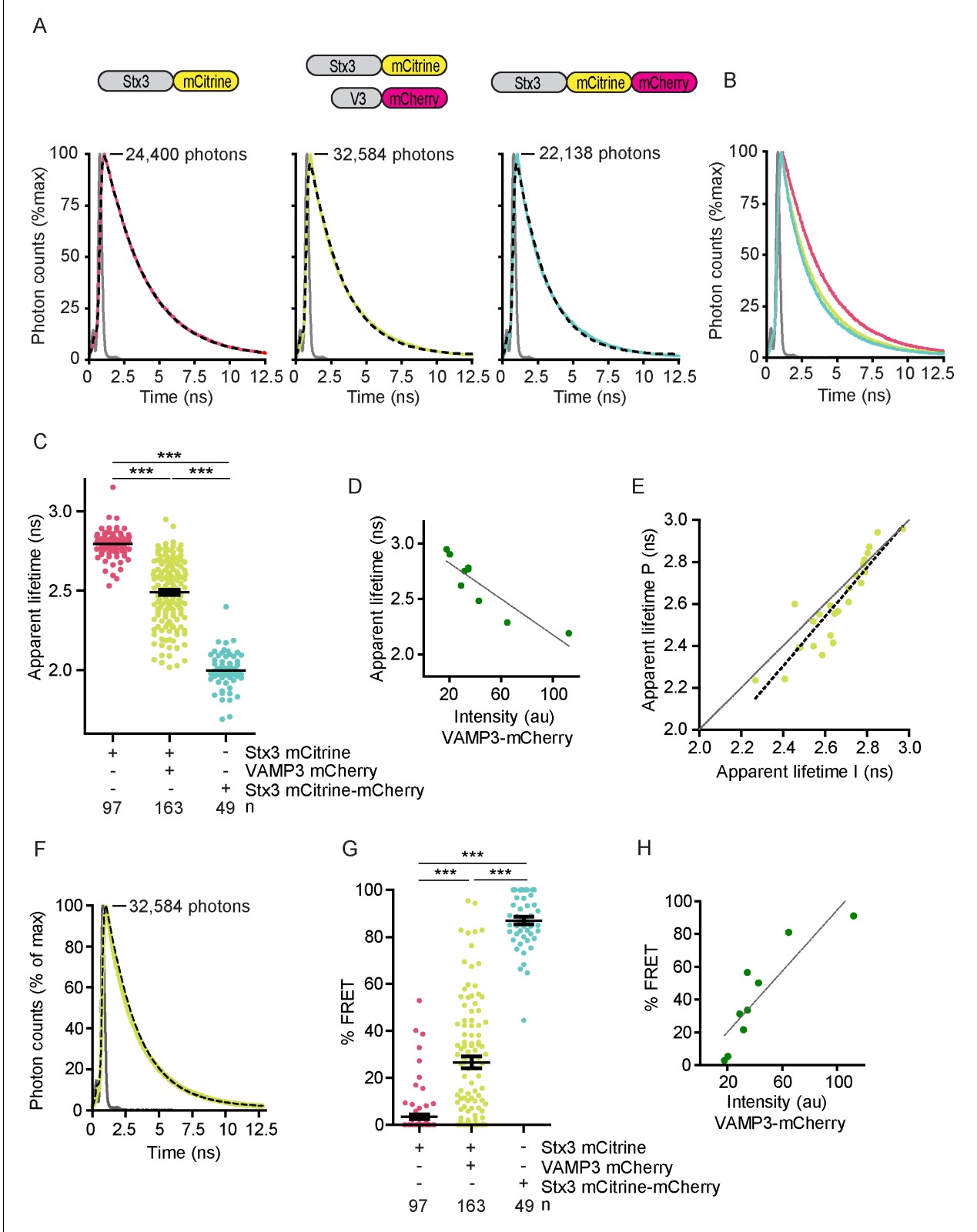

**Figure 2.** SNARE complex formation by whole-cell fluorescence lifetime measurements. (**A**) Representative whole-cell fluorescence lifetime histograms of dendritic cells expressing Stx3-mCitrine (red curves; left graph), Stx3-mCitrine with VAMP3-mCherry (green; middle graph), or Stx3 conjugated to both mCitrine and mCherry (Stx3-mCitrine-mCherry; cyan; right graph). Dashed lines: fits with mono-exponential decay functions convoluted with the instrument response function (IRF; gray). Graphs are normalized to the maximum photon counts (depicted in each graph). Apparent fluorescence

*Figure 2 continued on next page*

*Figure 2 continued*

lifetimes for Stx3-mCitrine: 2.90 ns; for Stx3-mCitrine with VAMP3-mCherry: 2.29 ns; for Stx3-mCitrine-mCherry: 2.05 ns. Shown with logarithmic scaling in *Figure 2—figure supplement 1A*. (B) Overlap of the fluorescence lifetime decay curves from panel *A* (logarithmic scaling in *Figure 2—figure supplement 1B*). (C) Whole-cell apparent fluorescence lifetimes for the conditions from panel *A*. Shown are individual cells pooled from at least 4 donors (mean ± SEM shown; one-way ANOVA with Bonferroni correction; n: number of cells; individual donors in *Figure 2—figure supplement 1D*). (D) Whole-cell apparent fluorescence lifetimes of Stx3-mCitrine as a function of the expression level of VAMP3-mCherry (by fluorescence intensities) of a representative donor (more donors in *Figure 2—figure supplement 1E*). Dashed line: linear regression ($\beta = -0.008$; $R^2 = 0.800$). (E) Apparent fluorescence lifetimes of dendritic cells expressing Stx3-mCitrine with VAMP3-mCherry at the peripheral region (P) *vs.* the internal region (I) of the imaged cell areas (*Figure 2—figure supplement 1F–G*). Individual cells from five donors are shown (grey curve: line of equality; black dashed curve: linear regression ($\beta = 1.174$; $R^2 = 0.821$)). Note that for most cells, the lifetime at the periphery is lower than at intracellular regions (Paired two-sided Student's *t*-test; p=0.0078). (F) Fluorescence lifetime histogram from panel *A* for a dendritic cell co-expressing Stx3-mCitrine with VAMP3-mCherry, but now fitted with a double-exponential decay function with the lifetimes of the slow (2.8 ns) and fast (2.0 ns) components fixed and convoluted with the IRF (gray curve). The percentage FRET (% FRET) was calculated as the amplitude of the fast component over the total amplitude and was 81% (logarithmic scaling in *Figure 2—figure supplement 2A*). (G) Same as panel C, but now fitted with double-exponential decay functions and % FRET shown. (H) Same as panel D, but now fitted with double-exponential decay functions and % FRET shown (more donors in *Figure 2—figure supplement 2B*). Dashed line: linear regression ($\beta = 0.927$; $R^2 = 0.771$).

The following figure supplements are available for figure 2:

**Figure supplement 1.** Fluorescence lifetime histograms fitted with mono-exponential decay functions and calibration of FLIM setup.

**Figure supplement 2.** Fluorescence lifetime histograms fitted with double-exponential decay functions.

free fit parameters since small errors in the lifetimes will influence the amplitudes of the two components and *vice versa*. The time constants of the slow and fast components of the double exponential decay function were fixed to the values of the donor only (i.e., no FRET; 2.79 ns) and of the tandem Stx3-mCitrine-mCherry construct (100% FRET; 2.0 ns). This allowed estimating the percentage of FRET from the amplitudes of the fast and slow components, which provides a measure for the fraction of Stx3-mCitrine in complex with VAMP3-mCherry (*Figure 2G*). Using this analysis, the percentage of complexed Stx3-mCitrine again correlated with the expression level of VAMP3-mCherry, and reached up to almost 100% at high expression levels (*Figure 2H*; *Figure 2—figure supplement 2B*). These data support our conclusion that complex formation of the donor fluorophore-labeled SNARE depends on the availability of acceptor fluorophore-labeled SNAREs. The observed fluorescence lifetimes of mCitrine not only depend on FRET, but will also be sensitive to other factors including the microenvironment, dipole orientation, and self-quenching of mCitrine within SNARE domains (*Zhu et al., 2015*). As a consequence, the observed fluorescence lifetimes in our samples may not be the same as in our control samples (i.e., donor only and tandem Stx3-mCitrine-mCherry). Fitting with mono-exponential decay functions does not require any a-priori knowledge of the fluorescence lifetimes. Moreover, correlation of the apparent lifetimes from mono-exponential fits with the amplitudes from the bi-exponential fits resulted in a clear linear correlation (*Figure 2—figure supplement 2C*; linear regression: $R^2 = 0.812$), demonstrating that these apparent lifetimes provide a reliable measure of the percentage FRET. We therefore analyzed the remainder of our FLIM data with mono-exponential decay functions.

## Validation of the FLIM assay for SNARE complex formation

We performed two sets of control experiments to confirm that FRET-FLIM is a reliable method to measure interactions between SNARE proteins in live cells. First, we generated fusion constructs of SNARE proteins with either the FK506 binding protein 12 (FKBP) or FKBP rapamycin binding (FRB) domain of mTOR fused to their N-termini. Dendritic cells were transfected with FRB-VAMP3-mCherry in combination with FKBP-Stx3-mCitrine or FKBP-Stx4-mCitrine (*Figure 3A*; *Figure 3—figure supplement 1A*). FKBP and FRB are rapamycin binding domains and the presence of this compound induces their dimerization (*Putyrski and Schultz, 2012*). Thus, we reasoned that the rapamycin-induced heteromerization of FKBP and FRB would force or stabilize SNARE association. Indeed, addition of rapamycin or the rapamycin analogue A/C Heterodimerizer resulted in a decrease of the apparent fluorescence lifetimes (*Figure 3A*).

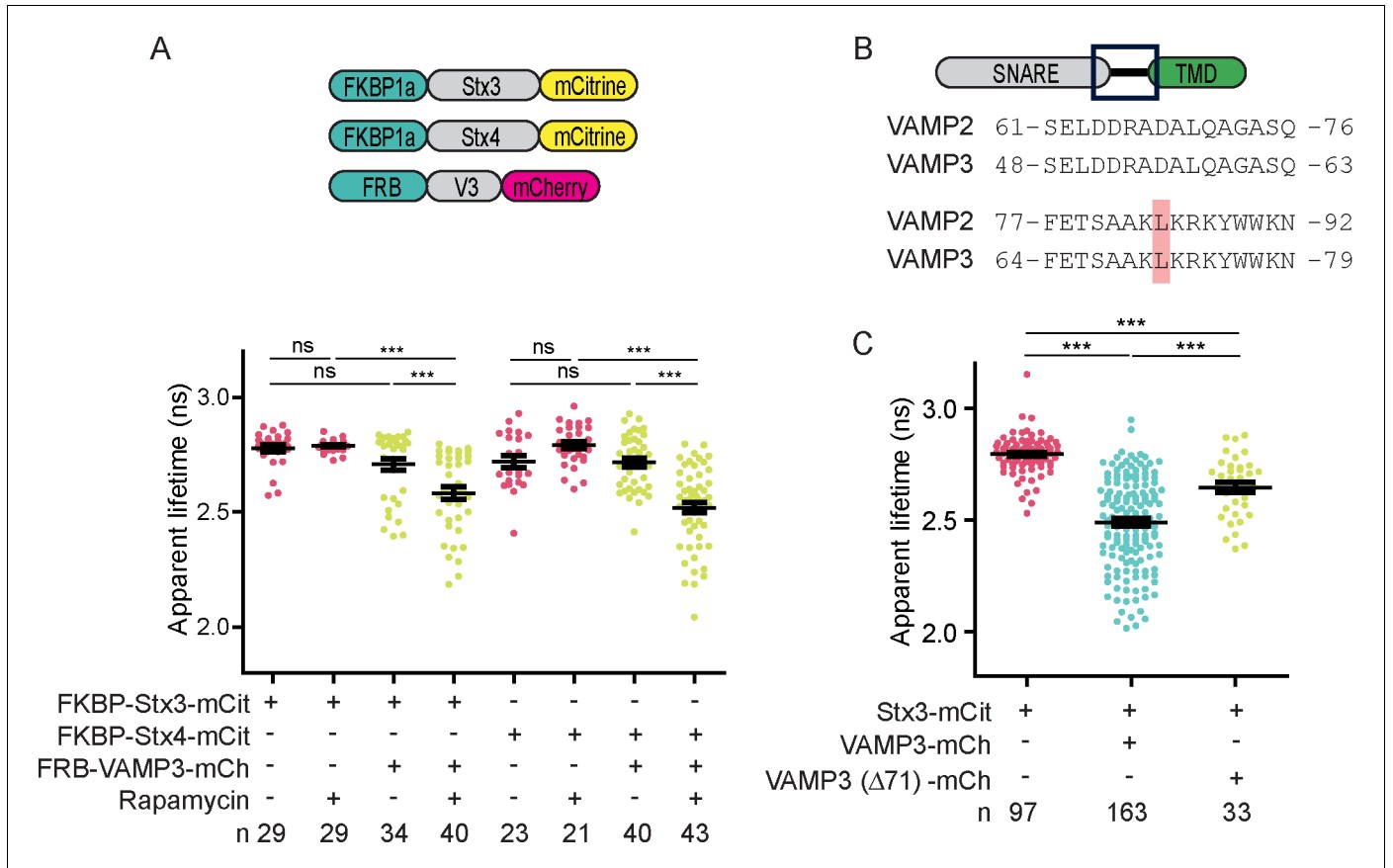

**Figure 3.** Forced interactions between SNAREs increases FRET while FRET is reduced with a fusion incompetent VAMP3 mutant. (**A**) Whole-cell apparent fluorescence lifetimes for dendritic cells expressing FKBP-Stx3-mCitrine or FKBP-Stx4-mCitrine together with FRB-VAMP3-mCherry and incubated in absence or presence of rapamycin or a rapamycin analogue. (**B**) Alignment of VAMP2 and VAMP3 (mouse sequences) showing 100% identity of the region containing leucines 84 (VAMP2) and 71 (VAMP3). (**C**) Whole-cell apparent fluorescence lifetimes for dendritic cells expressing Stx3-mCitrine with wild-type VAMP3-mCherry or a mutant lacking leucine 71 (VAMP3(Δ71)-mCherry). Shown in panels *A* and *C* are individual cells pooled from at least 4 donors (mean ± SEM shown; one-way ANOVA with Bonferroni correction; n: number of cells). Representative confocal and FLIM images are in *Figure 3—figure supplement 1*.

The following figure supplement is available for figure 3:

**Figure supplement 1.** FLIM images belonging to main *Figure 3*.

As a second approach to validate our FLIM method, we generated a mutant form of VAMP3-mCherry lacking leucine 71 (VAMP3(Δ71)) (*Figure 3B–C*; *Figure 3—figure supplement 1B*). This residue is located at the C-terminal end of the SNARE region which is identical to VAMP2 (*Figure 3B*). For VAMP2, deletion of leucine 84, homologous to leucine 71 of VAMP3, allows formation of a *trans*-SNARE complex but impairs fusion of membranes as progression to the *cis*-conformation cannot take place (*Hernandez et al., 2012*). For VAMP3, deletion of leucine 71 also impairs membrane fusion, as cells co-expressing Stx3-mCitrine with VAMP3(Δ71)-mCherry showed a significantly higher apparent fluorescence lifetime compared to cells expressing non-mutant VAMP3-mCherry (*Figure 3C*). These data confirm that FRET-FLIM allows detection of SNARE complexes.

## Comparison of different SNAREs involved in exocytosis

Next, we used our FRET-FLIM assay to compare complex formation between different SNARE proteins. Dendritic cells were transfected with Stx3-mCitrine or Stx4-mCitrine alone or in combination with VAMP3-mCherry or VAMP8-mCherry (*Figure 4A*; *Figure 4—figure supplement 1A*). Stx4-mCitrine mainly localized to the plasma membrane and only somewhat to intracellular compartments.

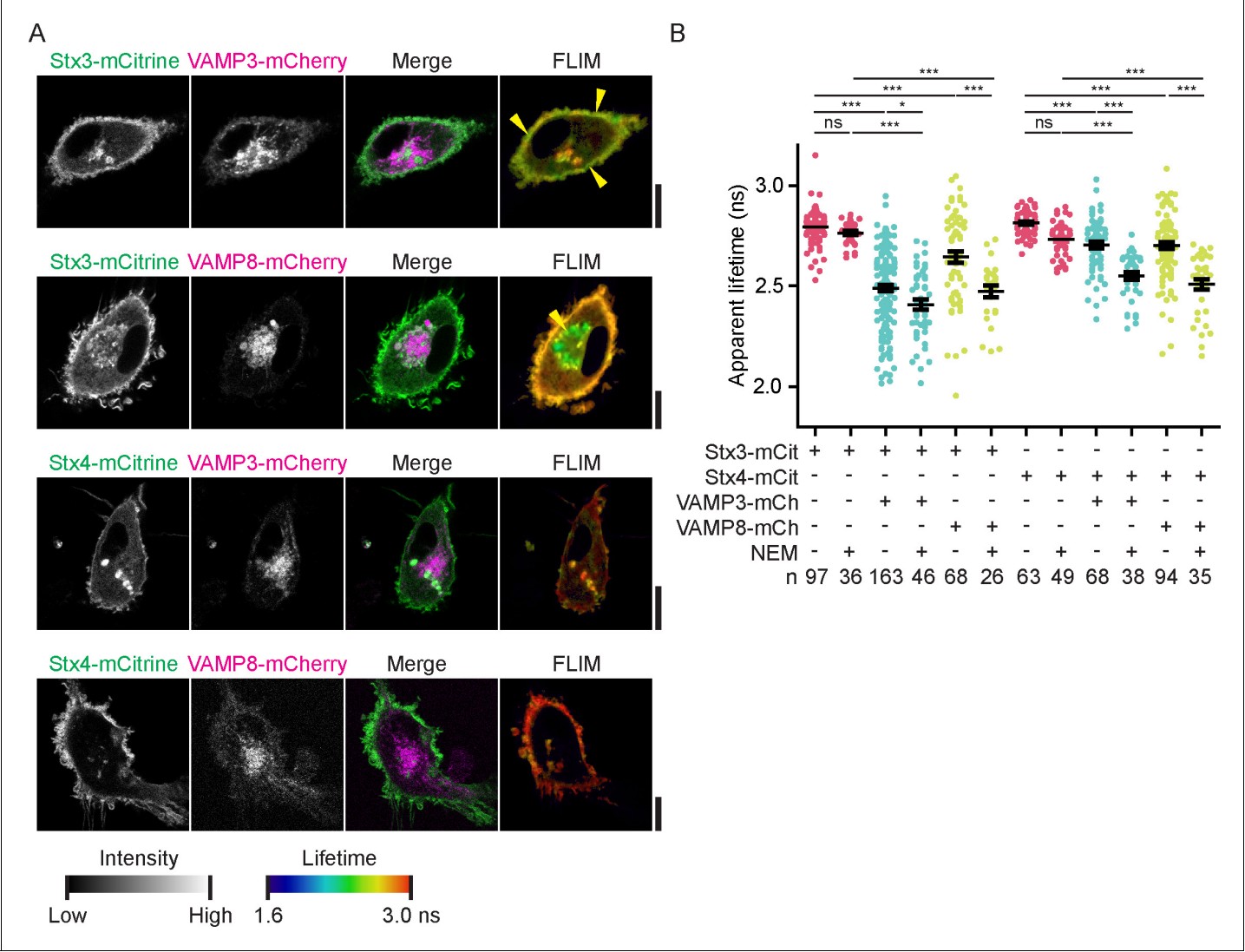

**Figure 4.** FRET-FLIM of Stx3 and Stx4 with VAMP3 and VAMP8. (**A**) Representative confocal microscopy (left) and convoluted FLIM (right) images of dendritic cells expressing Stx3-mCitrine or Stx4-mCitrine (green in merge) with VAMP3-mCherry or VAMP8-mCherry (magenta). The apparent fluorescence lifetimes of Stx3-mCitrine with VAMP3-mCherry were lowest at the cell membrane, whereas the lifetimes with VAMP8-mCherry were lowest at intracellular compartments (yellow arrowheads). Scale bars, 10 µm. Lifetime images are in *Figure 4—figure supplement 1A*. (**B**) Whole-cell apparent fluorescence lifetimes for the conditions from panel *A*, both in presence or absence of NEM. Shown are individual cells pooled from at least 3 donors (mean ± SEM shown; one-way ANOVA with Bonferroni correction; n: number of cells; individual donors in *Figure 4—figure supplement 1B*).

The following figure supplement is available for figure 4:

**Figure supplement 1.** Fluorescence lifetime images belonging to main *Figure 4*.

Similar to VAMP3-mCherry, the majority of VAMP8-mCherry was present in intracellular compartments (*Figure 4A*) and this corresponded to the localization of endogenous VAMP8 (*Figure 1—figure supplement 1B*). Whole-cell fluorescence lifetime analysis revealed interactions between all SNARE pairs tested, as reflected by reductions of the apparent fluorescence lifetime of mCitrine in the presence of mCherry-fused VAMP3 or VAMP8, although for Stx4 these reductions were relatively small (<100 ps) and not significant when analyzing donor-averaged lifetimes (*Figure 4B*; *Figure 4—figure supplement 1B*). FLIM imaging showed that whereas interactions of Stx3-mCitrine with VAMP3-mCherry mainly occurred at the plasma membrane, the interactions with VAMP8-mCherry predominantly occurred at intracellular compartments (*Figure 1C*, *Figure 4A*; *Figure 4—figure*

*supplement 1A*). As a positive control, we treated our cells with N-ethylmaleimide (NEM), which impairs the AAA-ATPase NSF responsible for dissociation of SNARE complexes. Addition of this compound traps all SNARE complexes in their *cis*-conformation (*Söllner et al., 1993*) and resulted in a trend showing further reductions of the apparent fluorescence lifetimes for all combinations of donor and acceptor SNAREs tested (*Figure 4B*; *Figure 4—figure supplement 1B*). Thus, our FLIM data resulted in two new observations of SNARE complex formation in unstimulated dendritic cells: (*i*) FRET at the plasma membrane of fluorescently-labeled VAMP3 with Stx3 was higher than with Stx4, suggesting more interactions of VAMP3 with Stx3 even though most Stx4 is present at the plasma membrane. (*ii*) FRET of fluorescently-labeled Stx3, but not Stx4, with VAMP3 was observed mainly at the plasma membrane while FRET with VAMP8 was found mainly at intracellular compartments, indicating that complex formation of these SNAREs occurred in different subcellular compartments.

## LPS stimulation promotes VAMP3 interaction with Stx4 at the plasma membrane

Activation of dendritic cells by inflammatory stimuli triggers the production and secretion of immunomodulatory cytokines and alters the surface receptors displayed on the cell membrane and this is accompanied by marked changes of the membrane trafficking machinery (*Collins et al., 2015*; *Stow et al., 2009*; *Stanley and Lacy, 2010*; *Lacy and Stow, 2011*; *Murray and Stow, 2014*; *Murray et al., 2005*; *Pagan et al., 2003*). We therefore wondered whether the SNARE interactions that we observed in unstimulated dendritic cells by FRET-FLIM would be altered by activation of the cells. We transfected dendritic cells with combinations of SNAREs and stimulated the cells overnight with the bacterial antigen LPS, a component from the outer cell membrane of gram-negative bacteria and a strong agonist of TLR4 (*Figure 5A*; *Figure 5—figure supplement 1A*). TLR4 stimulation triggers danger signaling in dendritic cells leading to cell maturation characterized by a strong increase in cytokine production. Although LPS stimulation can upregulate the expression of several SNAREs in immune cells (*Chiaruttini et al., 2016*; *Murray et al., 2005*; *Collins et al., 2014*; *Ho et al, 2009*; *Pagan et al., 2003*), the protein levels of both endogenous and overexpressed Stx3, Stx4, VAMP3 and VAMP8 in monocyte-derived dendritic cells were not altered by LPS (*Figure 5—figure supplement 1B–C*). Whole-cell FLIM analysis revealed that LPS activation of dendritic cells expressing Stx4-mCitrine in combination with VAMP3-mCherry led to a significant reduction of the apparent lifetime compared to the condition without LPS (*Figure 5B*; *Figure 5—figure supplement 1D*). In contrast, no significant changes were observed for Stx3-mCitrine or for VAMP8-mCherry. FLIM imaging demonstrated that the interactions of VAMP3-mCherry with Stx4-mCitrine mainly occurred at the plasma membrane (*Figure 5A*; *Figure 5—figure supplement 1A*). Because of the photobleaching we were unable to measure FLIM of the same cell pre- and post-LPS treatment. VAMP3 is known to mediate release of inflammatory cytokines by the murine macrophage cell line RAW264.7 (*Manderson et al., 2007*) and the human synovial sarcoma cell line SW982 (*Boddul et al., 2014*). VAMP3 is also required for the secretion of cytokines by LPS activated monocyte-derived dendritic cells, as siRNA knockdown of this SNARE impaired release of the cytokine IL-6, as shown by ELISA (*Figure 5C–D*; *Figure 5—figure supplement 1E*). These FLIM data support the conclusion that LPS activation of dendritic cells results in an increased complex formation of Stx4 with VAMP3 at the plasma membrane and that this promotes IL-6 secretion.

## Discussion

SNARE proteins are pivotal for membrane fusion and coordinate intracellular organellar trafficking, exocytosis and endocytosis (*Hong, 2005*; *Jahn and Scheller, 2006*). Although it is generally accepted that different intracellular trafficking routes are catalyzed by specific sets of SNARE proteins (*Collins et al., 2015*; *Stow et al., 2009*; *Stanley and Lacy, 2010*; *Lacy and Stow, 2011*; *Murray and Stow, 2014*), current microscopy techniques often cannot resolve distinct SNARE pairs for specific trafficking routes. In this study, we demonstrate that FRET-FLIM enables the assignment of SNARE partners to specific trafficking routes, as it allows visualization of SNARE pairing at subcellular resolution. Our data show that FLIM-FRET is a quantitative method that in principle allows to determine the fraction of SNAREs engaged in complex formation by fitting the photon counts with bi-exponential decay functions with fixed lifetimes (i.e., for no FRET (donor only) and 100% FRET (all

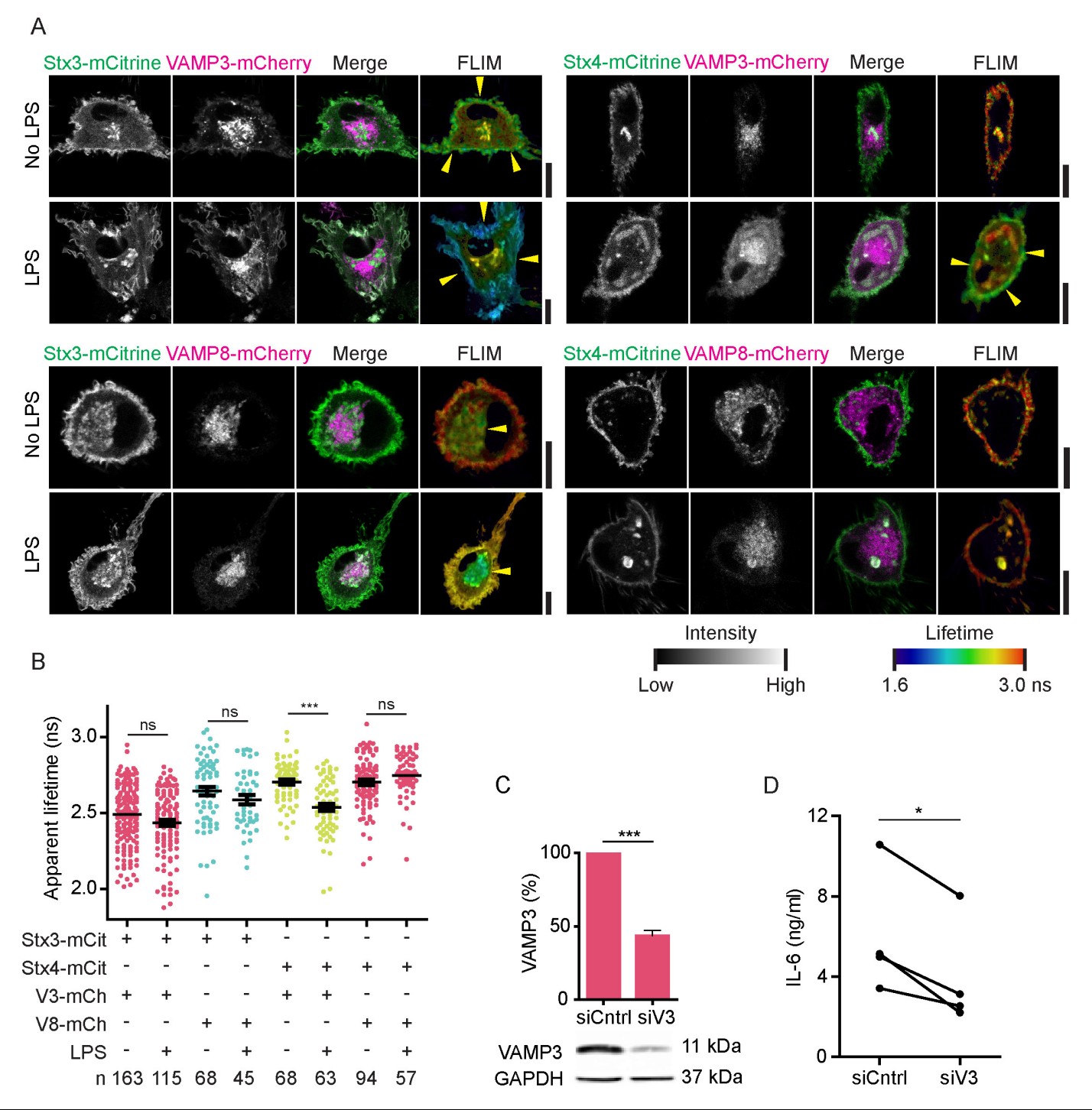

**Figure 5.** LPS activation of dendritic cells increases complex formation of Stx4 with VAMP3 at the plasma membrane. (**A**) Representative confocal microscopy (left) and convoluted FLIM (right) images of unstimulated or LPS-activated dendritic cells expressing Stx3-mCitrine or Stx4-mCitrine (green in merge) with VAMP3-mCherry or VAMP8-mCherry (magenta). Apparent fluorescence lifetimes of Stx3-mCitrine and Stx4-mCitrine with VAMP3-mCherry were lowest at the cell membrane (yellow arrowheads), whereas the lifetimes of Stx3-mCitrine with VAMP8-mCherry were lowest at intracellular compartments (yellow arrowheads). Scale bars, 10 μm. Lifetime images are in *Figure 5—figure supplement 1A*. (**B**) Whole-cell apparent fluorescence lifetimes for the conditions from panel *A* in presence or absence of LPS. Shown are individual cells pooled from at least 4 donors (mean ± SEM shown; one-way ANOVA with Bonferroni correction; n: number of cells; individual donors in *Figure 5—figure supplement 1D*). (**C**) Representative Western blot and quantification of siRNA knockdown of VAMP3 (siV3; p=0.0001, paired two-sided Student's t-test). siCntrl: non-targeting siRNA control. GAPDH:

*Figure 5 continued on next page*

Figure 5 continued

loading control. (D) IL-6 production after 24 hr LPS exposure by dendritic cells with VAMP3 knockdown (siV3) or siCntrl (individual donors shown; p=0.0199, paired two-sided Student's *t*-test; time traces shown in *Figure 5—figure supplement 1E*).

The following figure supplement is available for figure 5:

**Figure supplement 1.** Fluorescence lifetime images belonging to main *Figure 5*, expression levels of endogenous and overexpressed SNAREs upon LPS treatment and time traces of IL-6 secretion.

SNAREs interacting)). While the transient overexpression system used for our primary cells does not allow meaningful quantification of SNARE complexing, we expect that fusing chromosomal SNARE-encoding genes with fluorescent proteins using CRISPR/CAS9 in cell lines will enable to quantify endogenous SNARE interactions by FRET-FLIM.

We used FRET-FLIM to measure SNARE complex formation in human blood-derived dendritic cells. In unstimulated dendritic cells, the apparent lifetime of fluorescently labeled Stx3 with VAMP3 was lower at the periphery of the imaged cell areas, whereas the apparent lifetime with VAMP8 was lower at intracellular areas. This indicates that complex formation of Stx3 with VAMP3 mainly occurs at the plasma membrane, while interactions with VAMP8 predominantly occur at intracellular compartments. VAMP3 is primarily localized in early and recycling endosomes and VAMP8 locates predominately at late endosomal compartments (*Bajno et al., 2000*; *Manderson et al., 2007*; *Hong, 2005*; *Antonin et al., 2000*). Our data are therefore perfectly in line with the widely accepted notion that fusion of compartments of early/recycling endosomal nature with the plasma membrane is more abundant than of late endosomal compartments. Based on differences in apparent fluorescence lifetimes, we also conclude that the interactions of VAMP3 at the plasma membrane with Stx4 are lower than with Stx3 in unstimulated dendritic cells, even though the vast majority of Stx4 is present at the plasma membrane. This changes upon activation of the dendritic cells by the pathogenic stimulus LPS, which results in a decreased apparent lifetime of fluorescently labeled Stx4 with VAMP3 at the plasma membrane, whereas the apparent lifetimes of Stx3 remain unaltered. This result indicates that LPS triggers increased exocytosis by VAMP3 and Stx4.

VAMP3 is a recycling endosomal SNARE (*Bajno et al., 2000*; *Manderson et al., 2007*; *Hong, 2005*; *Murray et al., 2005*) and many cytokines traffic via recycling endosomes to the plasma membrane, including IL-6 and tumor necrosis factor alpha (TNFα) (*Manderson et al., 2007*; *Murray et al., 2005*). Indeed, our data show that siRNA knockdown of VAMP3 impairs IL-6 secretion by monocyte-derived dendritic cells as previously observed for RAW264.7 murine macrophages (*Manderson et al., 2007*) and SW982 human synovial sarcoma cells (*Boddul et al., 2014*). As supported by our FRET-FLIM data the increased complexing of VAMP3 with Stx4 is thus likely responsible for the increased secretion of IL-6 and other cytokines by activated dendritic cells. Interestingly, we observed that LPS increased FRET only of VAMP3 with Stx4 and not with Stx3. Stx3 and Stx4 have been well described in epithelial cells, where they catalyze apical and basolateral exocytosis respectively (*Low et al., 1996*; *ter Beest et al., 2005*; *Procino et al., 2008*), but why non-polarized cells such as dendritic cells express both syntaxins is not well understood. Our study provides a possible clue why this could be the case, as our data indicate that activation of the dendritic cells leads to a rerouting of intracellular trafficking with more VAMP3-containing compartments fusing with the plasma membrane by interacting with Stx4. Perhaps Stx3 catalyzes the constitutive cycling of early/recycling endosomes in resting cells and Stx4 offers the spare capacity to fulfil the additional secretory requirements upon dendritic cell activation needed for the secretion of massive amounts of cytokines.

What could cause the increased Stx4 complex formation with VAMP3 upon dendritic cell activation? Our data show that such increased SNARE complexing is not caused by increased levels of Stx4 or VAMP3. Possibly, the increased complexing might be regulated by direct phosphorylation of Stx4, VAMP3 and/or the Qb/c-SNARE SNAP23 (*Malmersjö et al., 2016*). SNAP23 was recently shown to be phosphorylated in an IκB-kinase two dependent fashion upon stimulation of mouse dendritic cells with LPS and this mediates the delivery of major histocompatibility complex (MHC) class I from recycling endosomes to phagosomes, presumably via Stx4 and VAMP3 or VAMP8 (*Nair-Gupta et al., 2014a*). Alternatively, or additionally, the increased complex formation could be

regulated by the Sec1/Munc18-protein Munc18c (STXBP3). Munc18c and Stx4 are well known to regulate GLUT4 trafficking in adipocytes and skeletal muscle cells (*Tellam et al., 1997*; *Hong, 2005*), and are also implied in insulin secretion from pancreatic beta cells (*Zhu et al., 2015*), secretion of dense-core granules, α-granules and lysosomes from human platelets (*Schraw et al., 2003*) and TNFα secretion from macrophages (*Pagan et al., 2003*). In any case, the mechanism must be specific for Stx4 and VAMP3, as our data show that FRET of Stx3 with VAMP3 and of Stx4 with VAMP8 are not altered by LPS-activation of dendritic cells. Overall, our study demonstrates the use of FRET-FLIM for the identification of the SNARE partners orchestrating specific organellar membrane trafficking steps. We expect this technique will allow deciphering the precise intracellular trafficking pathways of molecules important in health and disease, such as cytokines, receptors, hormones, metabolic enzymes and metabolites.

## Materials and methods

### Cells

Dendritic cells were derived from monocytes by culturing in the presence of IL-4 and GM-CSF for 6 days as described (*Dingjan et al., 2016*). Monocytes were isolated from blood of healthy individuals (informed consent and consent to publish obtained, approved by Sanquin ethical committee and according to Radboudumc institutional guidelines). Dendritic cells were transfected with plasmid DNA (overnight) or siRNA (for 48 hr; VAMP3 Stealth; HSS113848, HSS113849, HSS113850; Thermo Scientific, Waltham, MA) using a Neon Transfection system (Invitrogen, Carlsbad, CA) as described (*Dingjan et al., 2016*). To induce SNARE complex formation, NEM (400 µM; 10 min before imaging), rapamycin (Selleck Chemicals, Houston, TX) or the rapamycin analog A/C Heterodimerizer (Clontech, Mountain View, CA; #635057; 0.5 µM; 90 min before imaging) were added to the samples. LPS was used at 1 µg ml$^{-1}$ and incubated for 16 hr for FLIM measurements and for 4, 8 or 24 hr after which the supernatant was collected for IL-6 determination. IL-6 concentration was measured by ELISA (88-7066-88; Thermo Scientific).

### SNARE constructs

DNA coding for mCitrine was ordered as a synthetic gene and inserted in the BamHI/NotI sites of pEGFP-N1 (Clontech) yielding pmCitrine-N1. Stx3 and Stx4 (*ter Beest et al., 2005*) were inserted in the EcoRI/BamHI sites of pmCitrine-N1. VAMP3 and VAMP8 were inserted in the EcoRI/BamHI site of pmCherry-N1. The VAMP3(Δ71)-mCitrine mutant was generated by site directed mutagenesis. For FKBP-Stx4-mCitrine, synthetic DNA coding for human FKBP1A residues 1–108 with a serine-glycine linker (SGGGGSGGGGSGGGG) (Genscript, Piscataway, NJ) was subcloned in the restriction sites XhoI/EcoRI of Stx4-mCitrine. Stx4 was then replaced with Stx3 for FKBP-Stx3-mCitrine. For FRB-VAMP3-mCherry, residues 2,015–2114 of human mTOR with T2098L and a serine-glycine linker (Genscript) was subcloned in the XhoI/EcoRI restriction sites of the VAMP3-mCherry construct. For Stx3-mCitrine-mCherry, mCitrine with BamHI restriction sites on both sides was subcloned in the Stx3-mCherry construct.

### FLIM

FLIM was recorded with live (unfixed) dendritic cells on a Leica (Wetzlar, Germany) SP8 confocal microscope equipped with a 63 × 1.20 NA water immersion objective. Confocal images of the cells were recorded prior to each FLIM measurement. The image plane was selected at the height of the nucleus and was between 2–5 µm above the surface of the cover slips. Excitation was done with a pulsed white light laser (Leica; 80,000 MHz pulsing) operating at 516 nm. Emission from 521 to 565 nm was collected with a photomultiplier tube and processed by a PicoHarp 300 Time-Correlated Single Photon Counting (TCSPC) system (PicoQuant, Berlin, Germany). At least 50,000 photons with on average ~750,000 photons were recorded for each individual cell (*Figure 1—figure supplement 1C*). Samples were imaged in Live Cell Imaging Solution (Thermo Scientific) at 37°C. Photon traces in Picoquant PT3 format were used to construct FLIM images in Image Cytometry Standard (ICS) format using in-house developed PT32ICS conversion software (Membrane Trafficking Group, Radboudumc, Nijmegen, The Netherlands). For single cell FLIM, all photons were pooled for each individual cell and fitted with exponential decay functions convoluted with the IRF using OriginPro2016

(Originlab, Northampton, MA). Single pixel fitted FLIM images were generated for individual cells (with at least 1,000,000 photons per cell) using the TRI2 software (version 2.8.6.2; Gray institute, Oxford, UK) (*Barber et al., 2005*, *2009*) with monoexponential (Marquardt) fitting algorithm, $7 \times 7$ pixel circular binning and thresholding from 15% to 100% intensity. Phasor plots were generated using FIJI ImageJ (*Schneider et al., 2012*; *Schindelin et al., 2012*) with the Time Gated Phasor plugin from Spechron (developed by F. Fereidoni, UC Davis Medical Center).

## Immunofluorescence

Cells were plated and stained as described previously (*Dingjan et al., 2017*). For immunofluorescence labeling, primary antibodies rabbit anti-VAMP3 (Abcam, Cambridge, United Kingdom; #5789; 1:200 dilution (v/v)) or rabbit anti-VAMP8 (Synaptic Systems, Goettingen, Germany; #104–303; 1:200 dilution (v/v)) were used in combination with secondary antibody goat anti-rabbit conjugated with Alexa fluor 647 (Thermo Scientific; 1;400 dilution (v/v)). The cells were imaged on a Leica SP8 confocal microscope equipped with a $63 \times 1.20$ NA water immersion objective.

## Western blot

Dendritic cells were incubated for 16 hr in complete medium in the absence or presence of LPS (1 $\mu g\ ml^{-1}$). Cells were harvested and analyzed by Western blot. Primary antibodies used were a rabbit monoclonal against GAPDH (clone 14C10; Cell Signaling, Danvers, MA; #2118; 1:1000 dilution (v/v)), a rabbit polyclonal against Stx3 (*ter Beest et al., 2005*) (1:500 dilution (v/v)), a mouse monoclonal against Stx3 (clone 1–146; Millipore, Billerica, MA; MAB2258; 1:500 dilution (v/v)), a rabbit polyclonal against VAMP3 (Abcam #5789; 1:1000 dilution (v/v)), a mouse monoclonal against VAMP8 (Santa Cruz, Dallas, TX; sc-166820; 1:500 dilution (v/v)), a mouse IgG1 against Stx4 (Abcam #77037; 1:1000 dilution (v/v)) and a rabbit polyclonal against mCherry (Abcam #167453; 1:500 dilution (v/v)). Secondary antibodies were goat anti-rabbit or goat anti-mouse conjugated with IRDye 800 (LI-COR, Lincoln, NE; 1:5000 dilution (v/v)) or goat anti-rabbit conjugated with Alexa fluor 647 (Thermo Scientific; 1:500 dilution (v/v)).

## Statistics

Due to experimental restraints (time duration of the experiments and the viability of the cells), a maximum of 6 conditions a day could be measured. During single days, all relevant comparisons and appropriate controls were included. Cells from the same human subjects were used across the data displayed in *Figures 2–5*. For all the FLIM experiments, significant difference between two samples was assessed using one-way ANOVA with post-hoc Bonferroni correction for relevant pairs. Error bars indicate the SEM for at least >30 cells or for at least 3 donors. Western blots and ELISA results were analyzed by paired two-sided Student's $t$-test and error bars indicate the SEM for at least 3 donors. The level of statistical significance is represented by *$p \leq 0.05$, **$p \leq 0.01$, ***$p \leq 0.001$.

## Acknowledgements

This work was supported by a Starting Grant from the European Research Council (ERC) under the European Union's Seventh Framework Programme (Grant Agreement Number 336479). GvdB is funded by a Hypatia fellowship from the Radboud University Medical Center, a Career Development Award from the Human Frontier Science Program, the NWO Gravitation Programme 2013 (ICI-024.002.009), and a Vidi grant from the Netherlands Organization for Scientific Research (NWO-ALW VIDI 864.14.001).

## Additional information

### Funding

| Funder | Grant reference number | Author |
|---|---|---|
| Seventh Framework Programme | 336479 | Geert van den Bogaart |
| Nederlandse Organisatie voor Wetenschappelijk Onderzoek | NWO-ALW VIDI 864.14.001 | Geert van den Bogaart |

| Human Frontier Science Program | CDA-00022/2014 | Geert van den Bogaart |

The funders had no role in study design, data collection and interpretation, or the decision to submit the work for publication.

### Author contributions

DRJV, Conceptualization, Data curation, Formal analysis, Validation, Investigation, Visualization, Methodology, Writing—original draft, Writing—review and editing; NGM, Data curation, Formal analysis, Investigation, Writing—review and editing; MtB, Resources, Data curation, Investigation, Writing—review and editing; GvdB, Conceptualization, Software, Supervision, Funding acquisition, Methodology, Writing—original draft, Project administration, Writing—review and editing

### Author ORCIDs

Geert van den Bogaart, http://orcid.org/0000-0003-2180-6735

### Ethics

Human subjects: Monocytes were isolated from blood of healthy individuals (informed consent and consent to publish obtained, approved by Sanquin ethical committee and according to Radboudumc institutional guidelines).

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
