## [Decision Letter]

Thank you for submitting your article "Fluorescence Lifetime Imaging Microscopy reveals rerouting of SNARE trafficking driving dendritic cell activation" for consideration by *eLife*. Your article has been reviewed by two peer reviewers, and the evaluation has been overseen by a Reviewing Editor and Anna Akhmanova as the Senior Editor. The reviewers have opted to remain anonymous.

The reviewers have discussed the reviews with one another and the Reviewing Editor has drafted this decision to help you prepare a revised submission.

The manuscript by Verboogen et al. utilizes FRET-FLIM to study SNARE complex formation in dendritic cells and the role of VAMP3 and syntaxin 4 in LPS induced IL-6 release. Dynamic complexes formed between VAMP3 and Stx4 or Stx3 at the cell surface and VAMP8 complexes at intracellular sites are revealed. Finally the study concludes that LPS activation causes a switch to preferential formation of Stx4-VAMP3 complexes at the cell surface, which in turn support secretion of the IL-6 cytokine. This conclusion is novel in dendritic cells, in keeping with the functions of SNAREs in other immune cell types, but here it is reached on the basis of a single approach – albeit very elegant FLIM-FRET. The method that allows quantitative visualization of SNARE complexes with subcellular resolution is an important step forward. Overall the manuscript is interesting and the text well written. The figures are well presented and establish the approach (Figure 1–Figure 4) before addressing IL-6 exocytosis (Figure 5). There are several concerns that preclude publication in its current stage:

Figure 1. Authors state that there is decreased lifetime at the plasma membrane but do not attempt to quantify the difference. To make this statement (subsection 2SNAREs interactions in live cells visualized by FLIM”, second paragraph) they would need to quantify the lifetime at both the plasma membrane and the appropriate internal compartments across multiple examples.

In Figure 1 the authors note that Stx3-mCitrine localised to the plasma membrane and intracellular compartments. VAMP3-mCherry also located to the plasma membrane and predominantly in intracellular compartments. However, Stx3-mCitrine-mCherry appeared on the plasma membrane (mCitrine channel) and intracellular compartments (mCherry channel). Two concerns were raised. First, is the apparent localization of the VAMP down to the mCherry fusion? For example, all images in the mCherry channel in the manuscript show this accumulation. Second, numerous papers have used C-terminal fusions to VAMP proteins (e.g. pHlourins) and noted significant accumulation on the plasma membrane not observed for endogenous unfused VAMP. How does this mis-targetting impact the conclusions of the manuscript for example preferential pairing in the plasma membrane?

Figure 1 – The authors point out that the tandem dimer of mCitrine and mCherry does not 100% co-localize. They suggest that there might be differences in maturation. Another explanation is that the experimental design is not appropriate to measure FLIM at internal compartments such as lysosomes. As Vamp8-Syntaxin interaction is most likely at the late endosome/lysosome there will be a significant amount of quenching/degradation of mCitrine as compared to the relatively stable mCherry fluorophore that survives in the lysosomal compartment.

In Figure 2 FLIM is used by fitting a single decay to the entire image. The authors should show actual photon count numbers rather than normalized to 100%. This provides a clearer interpretation of this data for the reader. As shown in Figure 1 FRET lifetime is non-uniform in the cell. The authors should therefore use either a pixel by pixel fitting for all data with a bi-exponential decay or at least a bi-exponential fit to the whole cell data. The example data in Figure 2—figure supplement 1 shows that the fit is deviating at short lifetimes in the residuals.

Also, the authors note the large spread of lifetimes observed. Excluding the fitting issue above this is most likely due to the proportion of pixels reporting lifetime at the periphery versus the intracellular space. How was the imaging plane in the cell standardized? The authors suggest expression level as a potential issue, however, the number of plotted points in Figure 2 does not match the number of points in 2C. If concentration is the answer this would be better proved by using all data in 2C and examining correlation or using bi-exponential fits and examining amplitudes and lifetimes (preferably without fixing the short lifetime as it may not be the same as the positive control state).

Figure 4 and Figure 5 state that cells were used from 'at least 4 donors' or '3 donors' (were these donor numbers used as 'n' values for statistical purposes?). However, the cells appear to have been pooled to conduct the experiments instead of cells from each donor being measured separately. This would simply be a single mixed population rather than providing statistical replicates. If the authors have the separate data from each donor they need to include this and reassess their results.

In the text associated with Figure 4 the authors state "The interaction between Stx3-mCitrine and VAMP3-mCherry was stronger than for all other tested SNARE pairs". By stronger I presume they mean a shorter mean lifetime. However, this is misleading. The FRET energy transfer reported by the mono-exponential whole cell fit is a conflation of proportion of interacting molecules, proximity and dipole orientation averaged over the whole cell. The only conclusion that can be drawn is that the lifetime has changed. This issue also impacts on the subsection “Comparison of different SNAREs involved in exocytosis”, the Discussion and Abstract wherever the lifetime value is interpreted as a specific change in strength/number of interactions.

Figure 5: Given that FLIM is best suited to live cell imaging, Could the authors not show changes in FLIM in the same cell post LPS? This would give greater confidence in the observed changes pre- and post-LPS. In fact, it is not clear whether the study has used live or fixed cells, this needs to be stipulated in the methods.

Is Figure 5—figure supplement 1 endogenous or expressed isoform expression levels? There are no MW markers or text to indicate either way. Methods do not state if cells were transfected. As the experiment it controls for is examining over-expressed proteins this should be looking at over-expression level.

6) The IL-6 secretion data in Figure 5 is not very convincing in its current format. The experimental details need to be clarified (is this a 16 hour treatment with LPS followed by collection times of 4-24 hours (with or without LPS?) Secondly, the cytokine levels are currently expressed as percent of maximum. This is unconventional and does not convey direct information about the amount of cytokine secretion over the time course. Since the ELISA assay gives direct cytokine amounts, this is how the data should be conveyed (e.g. ng/ml) and ideally for each of the 3 donor cell lines +/- VAMP3.

---

## [Author Response]

*The manuscript by Verboogen et al. utilizes FRET-FLIM to study SNARE complex formation in dendritic cells and the role of VAMP3 and syntaxin 4 in LPS induced IL-6 release. Dynamic complexes formed between VAMP3 and Stx4 or Stx3 at the cell surface and VAMP8 complexes at intracellular sites are revealed. Finally the study concludes that LPS activation causes a switch to preferential formation of Stx4-VAMP3 complexes at the cell surface, which in turn support secretion of the IL-6 cytokine. This conclusion is novel in dendritic cells, in keeping with the functions of SNAREs in other immune cell types, but here it is reached on the basis of a single approach – albeit very elegant FLIM-FRET. The method that allows quantitative visualization of SNARE complexes with subcellular resolution is an important step forward. Overall the manuscript is interesting and the text well written. The figures are well presented and establish the approach (Figure 1–Figure 4) before addressing IL-6 exocytosis (Figure 5). There are several concerns that preclude publication in its current stage:*

*Figure 1. Authors state that there is decreased lifetime at the plasma membrane but do not attempt to quantify the difference. To make this statement (subsection 2SNAREs interactions in live cells visualized by FLIM”, second paragraph) they would need to quantify the lifetime at both the plasma membrane and the appropriate internal compartments across multiple examples.*

As suggested by the reviewers, we quantified the decrease in fluorescence lifetime at the plasma membrane area. We manually selected intracellular and peripheral regions in the FLIM images (new Figure 2—figure supplement 2). The photons from these regions were pooled for each individual cell and the fluorescence lifetime histograms were fitted with mono-exponential decay functions (new Figure 2—figure supplement 2). We then compared the fluorescence lifetimes at intracellular regions and the cellular periphery for multiple cells from 5 donors (new main Figure 2). The fluorescence lifetime was on average 55 ps lower at the cellular periphery compared to the intracellular region (P = 0.007; paired 2-sided t-test). However, the reduction was larger in cells with low fluorescence lifetimes (indicative of more SNARE complexing; linear regression, β = 1.174, R^[2]^ = 0.821). We now discuss this new analysis in the fourth paragraph of the subsection “SNAREs interactions in live cells visualized by FLIM”.

*In Figure 1 the authors note that Stx3-mCitrine localised to the plasma membrane and intracellular compartments. VAMP3-mCherry also located to the plasma membrane and predominantly in intracellular compartments. However, Stx3-mCitrine-mCherry appeared on the plasma membrane (mCitrine channel) and intracellular compartments (mCherry channel). Two concerns were raised. First, is the apparent localization of the VAMP down to the mCherry fusion? For example, all images in the mCherry channel in the manuscript show this accumulation. Second, numerous papers have used C-terminal fusions to VAMP proteins (e.g. pHlourins) and noted significant accumulation on the plasma membrane not observed for endogenous unfused VAMP. How does this mis-targetting impact the conclusions of the manuscript for example preferential pairing in the plasma membrane?*

We first determined to what extent fusing mCherry influences the intracellular localization of the SNARE proteins compared to mCitrine. We co-expressed VAMP3 and VAMP8 constructs fused to mCitrine with the same SNARE fused to mCherry (new Figure 1—figure supplement 1). We observed more accumulation of mCherry compared to mCitrine in an intracellular region juxtaposed to the nucleus, especially for VAMP8. As suggested by the reviewers (next point below), this accumulation of mCherry might be due to a higher resistance of mCherry to lysosomal degradation. It was not due to pH-quenching of mCitrine, as juxtanuclear accumulation of mCherry was also observed upon fixation of the cells (new Figure 1—figure supplement 1). As a consequence, we underestimate the amount of FRET in these juxtanuclear (lysosomal) regions and more stable YFP analogs need to be developed for this. However, we observed clear overlap of mCherry and mCitrine in more peripherally located cellular regions (Pearson correlation coefficient between 0.6–0.8 for the whole cell; new Figure 1—figure supplement 1) and believe it is justified to conclude that our FRET-FLIM method can report on SNARE interactions at these regions. These new data are now discussed in the third paragraph of the subsection “SNAREs interactions in live cells visualized by FLIM”.

To address the reviewers’ concern regarding the localization of the VAMP fusion constructs, we performed immunofluorescence microscopy of endogenous VAMP3 and VAMP8 in dendritic cells (new Figure 1—figure supplement 1). Both VAMP3 and VAMP8 were present mainly in intracellular compartments and less at the plasma membrane and this did not notably differ from the localization we observed for the VAMP fusion constructs. Please note that plasma membrane localization of endogenous VAMP3 and VAMP8 has been reported previously (e.g., Antonin, et al. (2000) Mol. Biol. Cell 11, 3289). We now discuss these new data in the second paragraph of the subsection “SNAREs interactions in live cells visualized by FLIM” and in the second paragraph of the Discussion.

*Figure 1 – The authors point out that the tandem dimer of mCitrine and mCherry does not 100% co-localize. They suggest that there might be differences in maturation. Another explanation is that the experimental design is not appropriate to measure FLIM at internal compartments such as lysosomes. As Vamp8-Syntaxin interaction is most likely at the late endosome/lysosome there will be a significant amount of quenching/degradation of mCitrine as compared to the relatively stable mCherry fluorophore that survives in the lysosomal compartment.*

As discussed in our second response above, we agree with the reviewer that it may very well be the case that mCitrine is more rapidly degraded compared to mCherry in degenerative compartments such as lysosomes. This limits the use of our FRET-FLIM method for measuring SNARE interactions in such compartments, and more stable YFP variants need to be developed for this. However, as the main conclusion from our manuscript concerns exocytosis at the plasma membrane, we believe the main conclusions of our study are not affected. We now discuss this caveat in the third paragraph of the subsection “SNAREs interactions in live cells visualized by FLIM”.

*In Figure 2 FLIM is used by fitting a single decay to the entire image. The authors should show actual photon count numbers rather than normalized to 100%. This provides a clearer interpretation of this data for the reader. As shown in Figure 1 FRET lifetime is non-uniform in the cell. The authors should therefore use either a pixel by pixel fitting for all data with a bi-exponential decay or at least a bi-exponential fit to the whole cell data. The example data in Figure 2—figure supplement 1 shows that the fit is deviating at short lifetimes in the residuals.*

*Also, the authors note the large spread of lifetimes observed. Excluding the fitting issue above this is most likely due to the proportion of pixels reporting lifetime at the periphery versus the intracellular space. How was the imaging plane in the cell standardized? The authors suggest expression level as a potential issue, however, the number of plotted points in Figure 2 does not match the number of points in 2C. If concentration is the answer this would be better proved by using all data in 2C and examining correlation or using bi-exponential fits and examining amplitudes and lifetimes (preferably without fixing the short lifetime as it may not be the same as the positive control state).*

As suggested by the reviewer, we now show the maximum number of photons in all lifetime histograms (main Figure 2; Figure 2—figure supplement 2, Figure 2—figure supplement 3A). However, we still normalize the *y*-scaling, because we want to compare conditions (with different photon counts) and present the IRF within the same graphs. As also suggested by the reviewers, we now fit all whole-cell lifetime histograms with biexponential decay functions (new main Figure 2). The deviations at short lifetimes (<2.5 ns) are mainly caused by imperfect fitting of the IRF. Technical issues such as drift of the laser pulsing or the timing of the detectors, reflections and/or (auto)fluorescence with fast kinetics may result in deviations at short lifetimes. However, please note that the deviations in our automated fitting are generally below 2%, which we believe is quite reasonable, and our experiments with the reference dye rhodamine B (Figure 2—figure supplement 2) indicate that this does not result in major changes of the fluorescence lifetimes. We have included the above discussion on the fit deviations in the fourth paragraph of the subsection “SNAREs interactions in live cells visualized by FLIM”.

As discussed above in our first response, we quantified the apparent fluorescence lifetimes at intracellular and peripheral regions of the imaged cells. Because primary blood-derived dendritic cells are very heterogeneous in size and morphology, it is difficult to standardize the image plane and we imaged at the height of the nucleus, which is between 2–5 µm above the surface of the cover slips. We now discuss this in the Results and Methods sections of our manuscript (subsection “SNAREs interactions in live cells visualized by FLIM”, second paragraph and subsection “FLIM”). As correctly noted by the reviewers, main Figure 2 (now main Figure 2) present data from a single representative donor. Our FLIM measurements were collected during a period of over 3 years, and because of differences in laser power, alignment of the excitation and emission optical paths, objective correction collar adjustment, etc., we cannot directly compare fluorescence intensities of data collected during this period. However, we observed the same trend for all donors, and now show three more representative donors in new Figure 2—figure supplement 2 and Figure 2—figure supplement 3B. Finally, as suggested by the reviewer, we examined the correlation of the amplitudes from biexponential fits with the apparent lifetime from mono-exponential fits and observed a clear linear correlation (new Figure 2—figure supplement 3C; β = -0.007, R^2^ = 0.867). In our biexponential fitting, we had to fix the lifetime components, as we did not have sufficient photon counts for an extra free fit parameter. Please note that fitting with biexponential decay functions with three free fit parameters (2 amplitudes and 1 lifetime) is extremely challenging, because small errors in the lifetime will influence the amplitudes and vice versa. Moreover, as the reviewers also note, lifetimes may not be the same as in the positive (and negative) control state(s), and could deviate because of multiple reasons (self-quenching, dipole orientation, variations in the microenvironment). We believe fitting with mono-exponential decay functions is warranted, because this does not require any a-priori knowledge of the lifetimes. To avoid any over-interpretation of our results, we present the lifetimes from these mono-exponential fits as *apparent* fluorescence lifetimes which are indicative of SNARE complex formation. We now provide this rationale for fitting the data with mono-exponential decay in the second paragraph of the subsection “SNAREs interactions in live cells visualized by FLIM”.

*Figure 4 and Figure 5 state that cells were used from 'at least 4 donors' or '3 donors' (were these donor numbers used as 'n' values for statistical purposes?). However, the cells appear to have been pooled to conduct the experiments instead of cells from each donor being measured separately. This would simply be a single mixed population rather than providing statistical replicates. If the authors have the separate data from each donor they need to include this and reassess their results.*

We provide the averaged data for the individual donors in the revised version of our manuscript (Figure 2—figure supplement 2, Figure 4—figure supplement 5B, Figure 5—figure supplement 6D). All effects that are important to support the conclusions of our manuscript are statistically significant both when analyzing single cells and donor averages. We choose to present the individual cell data in the main figures (with the donor-averaged data in the supplementary material), as FRET-FLIM is a single cell method and we want to show cellular heterogeneities.

*In the text associated with Figure 4 the authors state "The interaction between Stx3-mCitrine and VAMP3-mCherry was stronger than for all other tested SNARE pairs". By stronger I presume they mean a shorter mean lifetime. However, this is misleading. The FRET energy transfer reported by the mono-exponential whole cell fit is a conflation of proportion of interacting molecules, proximity and dipole orientation averaged over the whole cell. The only conclusion that can be drawn is that the lifetime has changed. This issue also impacts on the subsection “Comparison of different SNAREs involved in exocytosis”, the Discussion and Abstract wherever the lifetime value is interpreted as a specific change in strength/number of interactions.*

We revised the Abstract and the rest of the manuscript and now more carefully interpret our results. As requested by the reviewers, we now state that we observed reduced lifetimes, which is indicative of increased SNARE interactions.

*Figure 5: Given that FLIM is best suited to live cell imaging, Could the authors not show changes in FLIM in the same cell post LPS? This would give greater confidence in the observed changes pre- and post-LPS. In fact, it is not clear whether the study has used live or fixed cells, this needs to be stipulated in the methods.*

For all our FLIM recordings, we used live unfixed cells and we now emphasize this in our manuscript (Abstract, Introduction, Results and Methods sections). Unfortunately, we cannot do pre- and post-LPS of the same cell because of photobleaching. We now discuss this in our manuscript (subsection “LPS stimulation promotes VAMP3 interaction with Stx4 at the plasma membrane”).

*Is Figure 5—figure supplement 1 endogenous or expressed isoform expression levels? There are no MW markers or text to indicate either way. Methods do not state if cells were transfected. As the experiment it controls for is examining over-expressed proteins this should be looking at over-expression level.*

As suggested by the reviewers, we now provide the levels of both endogenous and overexpressed SNARE levels and indicate the molecular weights (Figure 5—figure supplement 6B, 6C). LPS treatment did not affect expression levels of both endogenous and overexpressed Stx3, Stx4, VAMP3 and VAMP8. We consider endogenous SNARE levels important, because endogenous SNAREs can compete with the overexpressed SNAREs and altered SNARE levels might thereby affect FLIM.

*6) The IL-6 secretion data in Figure 5 is not very convincing in its current format. The experimental details need to be clarified (is this a 16 hour treatment with LPS followed by collection times of 4-24 hours (with or without LPS?) Secondly, the cytokine levels are currently expressed as percent of maximum. This is unconventional and does not convey direct information about the amount of cytokine secretion over the time course. Since the ELISA assay gives direct cytokine amounts, this is how the data should be conveyed (e.g. ng/ml) and ideally for each of the 3 donor cell lines +/- VAMP3.*

We incubated the cells for 4–24h with LPS and collected IL-6 during this incubation period. We apologize for the unclarity and amended the Methods section (subsection “LPS stimulation promotes VAMP3 interaction with Stx4 at the plasma membrane”). As requested by the reviewers, we now show the absolute cytokine amounts and for each individual donor. Because of the large heterogeneity among donors, we show the 24 h time point in the main figure (main Figure 5) and the full time-traces for each individual donor in the supplementary material (Figure 5—figure supplement 6E).